# Unveiling the Behavior of an Endangered Facultative Cuprophyte *Coincya* Species in an Abandoned Copper Mine (Southeast Portugal)

**DOI:** 10.3390/plants13202847

**Published:** 2024-10-11

**Authors:** Ana Delaunay Caperta, Filipa Couchinho, Ana Cortinhas, Maria Manuela Abreu

**Affiliations:** LEAF—Linking Landscape, Environment, Agriculture and Food Research Center, Associate Laboratory TERRA, Instituto Superior de Agronomia, Universidade de Lisboa, Tapada da Ajuda, 1349-017 Lisboa, Portugal; couchinhofilipa@gmail.com (F.C.); analuciacortinhas@gmail.com (A.C.); manuelaabreu@isa.ulisboa.pt (M.M.A.)

**Keywords:** Aparis mine, *Coincya transtagana*, conservation, metallophytes, soil enzymatic activities, threatened species

## Abstract

Plant–soil interactions of endangered species with a high-priority conservation status are important to define in situ and ex situ conservation and restoration projects. The threatened endemic *Coincya transtagana*, thriving in the southwest of the Iberian Peninsula, can grow in metalliferous soils. The main goal of this study was to investigate the behavior of this species in soils rich in potentially toxic elements in the abandoned Aparis Cu mine. Soil samples were characterized for physicochemical properties and multielemental composition, as well as biological activity, through an analysis of enzymatic activities. Plant biomass was assessed, and multielemental analysis of the plants was also performed. The mine soils had slightly basic pH values and were non-saline and poor in mineral N-NH_4_, with medium-to-high organic matter concentration and medium cation-exchange capacity. In these soils, dehydrogenase had the highest activity, whereas protease had the lowest activity. The total concentrations of Cu (1.3–5.9 g/kg) and As (37.9–118 mg/kg) in soils were very high, and the available fraction of Cu in the soil also had high concentration values (49–491 mg/kg). Moreover, this study shows for the first time that *C. transtagana* had high uptake and translocation capacities from roots to shoots for Cu, Ni, and Cr. Although Cu in the plants’ aerial parts (40–286 mg/kg) was considered excessive/toxic, no signs of plant toxicity disorders or P uptake reduction were detected. This preliminary study revealed that *C. transtagana* is Cu-tolerant, and it could be used for phytoremediation of soils contaminated with potentially toxic elements, while also contributing to its conservation.

## 1. Introduction

Plant species that thrive in metal-rich mining soils have adapted to tolerate and sometimes accumulate high levels of metals/metalloids (metallophytes). The flora of such sites is of high conservation value [1,2] since the sites support highly distinctive plant communities that possess specific functional traits to adapt to extreme conditions [3]. The mining areas typically have high total concentrations (1- > 25 g/kg) of several potentially toxic elements (PTEs) such as Sb, As, Cd, Cu, Pb, Hg, W, and Zn [1,2,3]. In these areas, plant species with a widespread distribution or endemic ranges tolerating elevated concentrations of PTEs were identified [4,5,6,7]. These studies demonstrated that species with large distributions such as *Silene vulgaris* [4], *Thlaspi caerulescens* [5], *Biscutella laevigata* [6], and *Coincya monensis* [7] show differential metal accumulation in their tissues depending on the soil characteristics prevailing at the sites where they are found. By contrast, some plant species have a patchy distribution with limited geographic ranges and usually small population sizes such as the endemic *Agrostis durieui* in the northern Iberian Peninsula, which tolerates high levels of Pb concentration in its tissues [7], and the endemic taxa from serpentine soils in the California Floristic Province [8].

The high endemism often found in metal-rich habitats might be due to the low ability of metallophytes to colonize non-metalliferous habitats due to constitutive needs in metals [9]. Particularly, in soils naturally enriched with Cu that are rare on a global scale [3,10,11,12] or in Cu-enriched substrates due to mining activities, endemic species may occur (e.g., *Erica andevalensis* [12,13,14,15]). Although Cu excess is toxic for most species, Cu tolerance in metallophytes is thought to arise through the adaptation of mechanisms of basic metallic homeostasis [16]. Some of these species accumulate Cu and/or Co in their shoots and can be considered hyperaccumulators (>300 mg/kg [3,16]), such as in species found in the Copperbelt of Central Africa (Democratic Republic of Congo [16]). In abandoned Cu mines from the Portuguese sector of the Iberian Pyrite Belt (PIPB) like Lousal, São Domingos, and Chança, several endemic species are found, like *Erica australis* and *Cistus* spp. [17,18,19,20]. Another plant species found on disturbed soils in Cu mining soils is *Coincya transtagana* (Cout.) Clem.-Muñoz & Hern.-Berm. (Brassicaceae), an endemic species also observed in stony pastures and rocky slopes [21]. In mainland Portugal, assessment studies of this species revealed that there has been a steady decline in its area of occupation and the number of mature individuals due to agricultural expansion, as well as changes to the traditional way of managing cattle, being classified in the “Near Threatened” threat category of the IUCN [21].

Some studies have emphasized the importance of knowledge on plant–soil interactions for species conservation, particularly for endemic species that are afforded a high-priority status [11,22,23]. In this context, the understanding of relationships between mine soil and *C. transtagana* plants thriving in the abandoned area of the Aparis mine, formerly exploited for Cu ore, would help to develop conservation measures, as well as restoration projects, for this species. Through analysis of soil–plant interactions in *C. transtagana* in mine soils, the goals of this study were to (1) evaluate the physicochemical properties and multielemental concentrations of mine soils, as well as microbial activity through enzyme activities; (2) determine plants’ chemical multielement concentrations; and (3) assess plant behavior through soil-to-plant coefficients. Although this is a coarse-scale analysis, this constitutes the first soil–plant interactions study for this endangered species and provides the basis for subsequent fine-scale studies on the conservation of this species.

## 2. Results and Discussion

### 2.1. Soils Characterization

In the mine area, the soils classified as Spolic Technosol Toxic, were incipient and had developed on different mine wastes and host rocks [24]. The soils’ properties are shown in Table 1. The soils were moderately basic, except for soil L5, which was slightly acid to neutral. They were considered non-saline (EC values < 0.4 mS/cm), excluding soil L1, which was slightly saline since it was developed on fine materials resulting from ore-processing techniques (EC = 4.06 mS/cm). The presence of carbonates was found in all of the samples, which is in agreement with the geology of the mine area, where quartz veins and veins associated with carbonates occur. The samples presented high concentrations of both organic carbon (Corg) and total nitrogen (N_total_), except for soil L1. The N-NH_4_ was below the detection limit, but N-NO_3_ values were relatively high for this soil type. The medium value for the C:N ratio (10.5) indicated a tendency for fast Corg mineralization [25].

Concentrations of extractable P in soils were variable, classified into different fertility classes like low (soils L1 and L4), medium (soils L3 and L5), and high (soil L2) [26]. The extractable K values also varied among soils, ranging from medium (L1 and L5 soils) to high (L2 and L3 soils) and very high (L4 soil) fertility classes [27].

Globally, the cation-exchange capacity (CEC) of the soils was moderate except for soil L1, which was very low (<5 cmol_c_/kg; [27]). The concentrations of exchangeable cations Ca and Mg were, in general, medium to high, but they were low in soil L1. The values for the exchangeable K were low for soils L1 and L5 and medium for the remaining soils. The exchangeable Na concentration was very low for all samples.

The micronutrient concentration was very high for Fe and Cu [27], attaining 2 g Cu/kg in soils L3 and L5, while Zn and Mn concentrations in soils L1 and L2 had medium and high values, respectively, and high to very high values for the other studied soils. The macronutrients presented values that are within the same range of the same cations in the exchangeable complex.

The multielemental (pseudototal) soil composition revealed that PTEs (Table 2) were below the maximum allowed values (MAVs) [28], except for As and Cu, which exceeded these values (11 mg/kg and 140 mg/kg, respectively), and Co in soils L3, L4, and L5. Soil L1, which developed on wastes from ore processing, had the lowest PTE concentration values, except for As and Mn. The concentrations of other PTEs like Cd, Cr, Mo, Sb, and Sn were low and below the MAVs. Regarding the available fraction, the values of those elements, as well as of Pb, were mostly below the detection limit of the analytical device used. Even exceeding the MAVs, when considering the pseudototal values, the available fraction of As and Cu concentrations in the soil corresponded to percentages lower than 10%, excluding As in soil L1, which attained 16.5% but whose concentration (4.1 mg/kg) was lower than the MAV.

In Aparis soils, Cu had higher values than abandoned Mociços and Miguel Vacas Cu mines also located in the Ossa Morena region [29,30] and on average 56- and 53-times-higher values than in other abandoned PIPB Cu mines like Lousal, São Domingos, and Chança [18,20]. This is probably related to the host rocks from which the soils developed, as well as Cu recovery by ore processing. However, the values for As, Mn, and Zn in Aparis soils were lower than in the above-mentioned PIPB mine soils but higher than in Mociços mine. Cobalt values also exceeded the MAVs in two Aparis soils but were within the range determined in the Mociços mining area [29], being relatively lower than that found in São Domingos mine soils [31].

### 2.2. Enzymatic Activities

Enzymatic activities (Figure 1) were used as biological indicators to assess the soils’ biological activities, since these enzymes mediate and catalyze several soil biochemical and nutrient-cycling processes involved in soil functions [32,33]. General microorganism activity was low in soils L1 (2.54 ± 0.27 μg TPF.g^−1^) and L5 (1.99 ±0.38 μg TPF.g^−1^), as evaluated by dehydrogenase activity, probably associated with low Corg concentration values. Organic carbon is the main source of microbial activity and enzymatic activities, which increase in response to the increase in soil microbial populations and their activity [32,33,34].

The lowest values of all the enzymatic activities were obtained in soil L1, which presented the highest EC value and calcium carbonate content. Similar findings were also obtained for β-glucosidase, acid phosphatase, and protease activities in soils from Egypt with different calcium carbonates levels and EC [35]. Acid phosphatase, β-glucosidase, and cellulase activities followed the same tendency of dehydrogenase activity, notably having the lowest activity in both soil L1 and soil L5 (Figure 1). These last two enzymes are stimulated by soil Corg content [36]. β-glucosidase and cellulase are carbon-cycle enzymes, having a major role in soil organic matter and plant residue degradation [37]. In this current study, the results revealed that β-glucosidase activity, at least, was the highest in soils presenting greater Corg content (L1, L2, and L3). Compared to other PIPB Cu mines, in this study, dehydrogenase, acid phosphatase, and β-glucosidase activities were higher than in São Domingos mine soils [38]. Both N-cycling enzymes, protease and urease [39], exhibited the lowest activity in soil L1, which had the lowest N_total_ and N-NO_3_ content (0.37 g/kg and 2.2 mg/kg, respectively). Urease activity was higher in the São Domingos mine than in the studied mine soils [38]. Sulfatase enzymes are very important in S mineralization and are involved in the S cycle of inorganic nutrients and metals in mine soils [40]. Sulfatase activity was significantly higher in soils L2 and L4 than in the remaining soils, which could be attributed to the lower Cu content in the soils’ available fraction [41]. In this latter study, it was also revealed that increasing Cu application to soil affected dehydrogenase, acid phosphatase, and sulfatase activities, which is in agreement with our findings.

### 2.3. Plant Analysis

#### 2.3.1. Flowers Characterization

There were few plants (<50) with a dispersed distribution in soils L1-L2-L3 in contrast to soils L4 and L5 (~200). The plants showed typical *C. transtagana* diagnostic characteristics, although some of them presented slightly different corolla patterns, namely a corolla with yellow petals without a halo or with yellow petals forming a white halo or a violet halo (Figure 2).

Although *C. transtagana* usually has flowers with white or pale-yellow petals with purple veins [21], plants with flowers with white petals were not found in this study. By contrast, it was observed that in soils L1 (n = 18) and L2 (n = 15), all of the sampled individuals had yellow petals, whereas in L3 most individuals had a corolla forming a white halo (n = 13) and a few had a violet halo (n = 3 out of 16). In soil L4 (n = 39), individuals with all three corolla patterns were detected, while in soil L5 (n = 39) only individuals with a corolla forming a white halo were observed. Flower color, pattern, shape, and reward relate to physiological adaptations to stress, as found in the *Mimulus guttatus* (Phrymaceae) complex adapted to Cu mine tailings and soils developed on serpentine [42].

#### 2.3.2. *Coincya transtagana* Behavior

Knowledge on *C. transtagana* behavior, thriving in metalliferous mine soils, is not only important for its conservation but could also be a biological resource for technological applications like the phytoremediation of contaminated soils [11,22,23].

*Coincya transtagana* plants’ aerial part (AP, stems, leaves and flowers) and roots biomass were greater in L2 site, which soil presented the highest values of extractable P and total N (Table 3), particularly the mineral N fraction (N-NO_3_ = 13 mg/kg) easily uptake by plants. The soil L2 had the lowest values of PTEs in the available fraction especially Cu, with a value between three and ten times lower than in the other soils (Table 2). The concentrations of macronutrients Ca, Mg, K, P and S (Table 3) were generally considered adequate for a healthy plant vegetative development [43], being an intensive (K, P, S) or a strong (Ca, Mg) uptake from the soil. This is confirmed by BAC values ranging from 10 to > 100 and 1 to 10, respectively. As expected, these nutrients were efficiently translocated from roots to shoots (TransferC > 1).

This species behavior in relation to As was not clear, since two samples (L1 and L3) presented As values in AP that were considered excessive or toxic (5–20 mg As/kg; [44]), whereas in the other samples the values determined in the plant digestion solution were lower than the detection limit of the apparatus (<3 mg/kg in plant dry mass). Although As concentrations in the soils’ available fraction was similar in soils L3-L4-L5 (1.1–1.7 mg/kg of soil) and even higher in soil L2 (2.7 mg/kg of soil), plants developing in such soils have different behaviors concerning As uptake.

The studied species showed an intense absorption capacity of Cr (BAC: 12.6–30.4, Table 4), although this element availability for plant uptake was very low (soil L1 0.3; other soils <0.2 mg/kg Table 2). The Cr values in the plants’ AP (Table 3) exceeded the normal range for plants in general (0.1–5 mg/kg; [44]), being in the excessive or toxic range (5–30 mg/kg; [44]). Despite high Cr concentrations, this species appeared to be tolerant to this element, since the plants had no visible symptoms of toxicity as chlorosis [44,45] and were not Cr hyperaccumulators (AP < 1000 mg/kg; [46]). *Coincya transtagana* translocated this element to AP (TranslC > 1) as opposed to other plant species [44,47]. The high soil–plant Cr transfer can be attributed to the soil pH (neutral to alkaline, Table 1) and the species of Cr in Aparis soils. In these soils, Cr probably is in, as it is the most mobile species and easily absorbed by plants [44,47]. On the other hand, these soils contain relatively high Mn concentrations (Table 2) that promote the oxidation of Cr(III) to Cr(VI) [44,48], which is more mobile and preferentially uptaken.

*Coincya transtagana* showed a strong absorption capacity for Fe (BAC 1–10), which was related to the high Fe concentration in Aparis soils (pseudototal: 27–43 g/kg; available fraction: 63–168 mg/kg). Values like those determined in the studied species were also reported for other plant species thriving in soils with high available Fe content [44,49]. The studied species also had higher AP Fe levels (e.g., L1 plants—5.87 g/kg) than *Lavandula stoechas* subsp. *luisieri*, *Origanum vulgare* subsp. *virens*, and *Calamintha nepeta* subsp. nepeta grown in Fe-Mn Rosalgar mine soils [50].

Lead concentration found in *C. transtagana* plants was low (5–10 mg/kg; [44]) and below concentrations found in other species from genera *Cistus*, *Lavandula*, and *Erica* thriving in Cu mine soils [18,19,51] and in *Coincya monensis* (29.41 mg/kg) growing on Pb-Zn mining wastes in the Cantabrian range (Spain) [7]. Generally, Mn concentration in plants’ AP was within the normal range (30–300 mg/kg; [44]), with a medium absorption capacity (BAC 0.3–0.9, Table 4). In comparison with other species growing in PIPB mine soils like *Cistus* sp. [18] or species from Pb mines such as *Acacia melanoxilon*, *Cistus inflatus*, and *Erica arborea* [52], *C. transtagana* had low Mn concentration in plants’ AP.

Regarding other micronutrients like Ni, Mo, Cu, and Zn, Ni content in plants’ AP was adequate (1–10 mg/kg; [43]) and within the same range found in other plant species thriving in S. Domingos mine soils [31]. However, L1 plants presented excessive or toxic Ni values [44], with a TranslC > 1 (Table 4). The Mo concentration in plants’ AP was above the range considered sufficient or normal (0.2–5 mg/kg; [44]) in three samples (L1, L2, and L5) but not in the excessive or toxic range (10–50 mg/kg; [42]). In Aparis soils, the pseudototal concentration of Mo (0.6–0.8 mg/kg) was within the range of values given by the European baseline topsoil (median 0.62 mg/kg, [53]), but this element’s available fraction was below the detection limit (<0.05 mg/kg). Therefore, *C. transtagana* showed a high capacity for Mo uptake and its translocation from the roots to the shoots (TranslC > 1, Table 4). Copper content in plants’ AP was high, within the range considered excessive/toxic (20–100 mg/kg; [44]). However, no signs of toxicity disorders were observed such as Fe chlorosis or P uptake reduction. The P content (Table 3) in plants’ AP was compatible with a good vegetative development, and Fe concentrations were above the normal values. These findings revealed that there was no antagonism between Cu-Fe and Cu-P during absorption and translocation [44]. Despite the high Cu concentration in the soils’ pseudototal and available fractions (Table 2), the Cu absorption capacity of *C. transtagana* was classified as medium, and the TransferC was very low (0.01–0.31) (Table 4). The high Cu tolerance in this species can be related to the high Ca concentrations in plants [54] or to Si concentration both in AP (468–635 mg/kg dry weight) and the soil’s available fraction (43–57 mg/kg). Silicon enhances tolerance to some potential toxic metals such as Al, Cd, Pb, Cr, and Cu, but its effect on metal uptake and translocation depends on the plant species and genotype [55,56,57]. In comparison with the *Erica*, *Lavandula*, and *Cistus* spp. growing in Cu mine soils [18,19,31,41,58], the Cu levels quantified in *C. transtagana* were much higher, but this plant species is not a Cu accumulator (TransferC < 1). The Zn concentration in *C. transtagana* AP was within the normal range for plants (27–150 mg/kg; [44]), with a strong-to-intensive Zn absorption capacity (BAC 6.8–21.3), and a medium value for element translocation from roots to shoots (TranslC 0.8–1.7). In *Cistus* and *Lavandula* spp. grown in abandoned Cu mine soils [18,51], the Zn TranslC was much higher (TranslC 2.5–5.3). Remarkably, other *Coincya* species like *C. monensis* had shoot Zn concentrations (3.4 g/kg) in the hyperaccumulator range and a very high Cd bioaccumulation factor in Pb-Zn mine spoils [7].

## 3. Materials and Methods

### 3.1. Study Species

*Coincya transtagana* is an annual plant that typically shows flowers with white or pale-yellow petals (14–18 × 3–4.5 mm) with purple veins and strongly curved and waxy fruits that can be dehisced or undehisced [59] (Figure 3). This species is endemic to the Iberian Peninsula, scattered in the south and southeast of Alentejo and northern Algarve in Portugal and in southwest Spain, where it occupies stony pastures, scrubland clearings and rocky slopes and in disturbed soils of mining areas [59]. Risk assessment studies on this species in continental Portugal classified it as critically endangered following IUCN criteria [21], and it was considered endangered in Extremadura, in southwest Spain [60,61].

In the Aparis mine area, *C. transtagana* forms communities with other rare species like *Armeria linkiana* Nieto Fel. and *Prolongoa hispanica* G. López & C.E. Jarvis, in addition to common dandelions (e.g., *Coleostephus myconis* (L.) Rchb.f.; *Crepis capillaris* (L.) Wallr.), marigolds (e.g., *Chamaemelum mixtum* (L.) All.; *Chamaemelum fuscatum* (Brot.) Vasc.), spike thistle *Galactites tomentosus* Moench, and *Echium plantagineum* L. [62].

### 3.2. Site Characterization

The Aparis mine was one of the most important mining areas in Barrancos (southeastern Portugal, 38°7′25″ N, 7°4′49″ W), being operational between 1889 and 1932 and in beginning of the 70s, closing in 1975 [63,64]. Ore was extracted by underground mining, with mining shafts reaching a depth of 150 m.

This mine, included in the morphostructural unit of the Ossa Morena Zone [65], is in the Sousel-Barrancos Belt, which extends along the Estremoz–Barrancos lithostratigraphic sector that includes a high number of cupriferous epigenetic mineralizing systems [65,66]. The mineralization is of a vein type, related to a set of fault zones, and consisting of a main vein and other secondary veins, with most of them made up of different generations of milky quartz and carbonates. The primary mineralization at Aparis mainly includes different generations of chalcopyrite related to carbonates (siderite and ankerite), iron sulfides (pyrite, marcasite, arsenopyrite, sphalerite, pyrrhotite, and galena), and sulfosalts (tetrahedrite–tennantite) that occur in fractured domains of the veins [65,66]. Supergene enrichments are common, leading to the development of mineral associations consisting of iron oxyhydroxides, malachite/azurite, cuprite, libethenite, chrysocolla, and covellite [66].

The climate of the area is classified as Csa according to the Köppen classification, with hot and dry summers and wet and moderately cold winters. The total average precipitation is 555 mm [67].

The mine area is surrounded by a natural pasture and nearby oleanders (*Nerium oleander* L.) and reeds (*Carex* sp.; *Juncus* sp.) thriving in riverbanks of the creek in the mining area besides holm oaks (*Quercus rotundifolia* Lam.). Herds of sheep and cattle grazing are visible in this area.

### 3.3. Soil and Plant Sampling, and Characterization

Due to the mining activity and to its direct impact on the areas of present *C. transtagana* populations, habitat fragmentation was observed. In the mine area, five subpopulations were identified, and five sampling sites (L1–L5, 2–3 m^2^ each site) were chosen according to the number of individuals found. In each site, a composite soil sample was obtained from the homogenate of three subsamples (total ≈ 3 kg) collected (0–5 cm of depth) in the surrounding area of the radicular system of the plants.

The soil samples were air-dried, and the fraction <2 mm was used to determine pH and electrical conductivity (EC) in water (1:2.5 *m*/*V*), total nitrogen (Kjedahl method), nitric and ammoniacal nitrogen (molecular absorption spectrophotometry in a segmented flow autoanalyzer, preceded by Micro Kjeldahl acid digestion for ammonia nitrogen) [68], extractable P (Olsen method [26]), extractable K (Egner–Riehm method—0.04 mol/dm^3^ ammonium lactate extractant agent acidified by hydrochloric acid at pH 3.5–3.7; [69]), cation-exchange capacity by 1 mol/dm^3^ ammonium acetate buffered to pH 7, as described in [70], macro and micronutrients quantified by atomic absorption spectrophotometry after extraction with an aqueous solution of ammonium acetate 0.5 mol/dm^3^-acetic acid 0.5 mol/dm^3^-EDTA 0.02 mol/dm^3^ at pH 4.65 [71], and organic C (Sauerland method). The available fraction of the elements in the soil was determined by ICP-OES after soil extraction with an aqueous solution (3:20 *m*/*V*) composed of a mixture of organic acids (acetic acid + lactic acid + citric acid + malic acid + formic acid at 10 mmol/dm^3^) for 16 h of agitation to simulate rhizosphere conditions [72] and a multielemental analysis by ICP-MS after aqua regia digestion [73].

The soils’ enzymatic activities were evaluated in all of the samples, which were stored at 4 °C after collection until analysis and sieved at 2 mm for β-glucosidase (based on the colorimetric determination of p-nitrophenol generated after soil incubation (1 h at 37 °C) with β-D-glucopyranoxide in a buffered medium at pH 6) [74]; cellulase (based on the quantification of glucose release after incubation with a microcrystalline cellulose substrate (Avicel) in a buffered solution at pH 5.5 and 40 °C for 16 h) [75]; dehydrogenase (method based on the reduction of 2,3,5-triphenyltetrazolium chloride to triphenylformazan after incubation with the soil at 25 °C in the dark for 16 h) [74]; phosphatase (based on the spectrophotometry of p-nitrophenol released during soil incubation (37 °C for 1 h) with a buffered p-nitrophenylphosphate solution) [76]; protease (method based on colorimetric determination via the Folin reaction of the released peptides soluble in trichloroacetic acid after soil incubation with casein for 2 h at 50 °C) [77]; sulfatase (based on the colorimetric determination of p-nitrophenol generated after soil incubation (1 h at 37 °C) with p-nitrophenyl sulfate in a sodium acetate buffer medium at pH 5.8) [78]; and urease (the method is based on the colorimetric determination of ammonium released after soil incubation with an urea solution for two hours at 37 °C) [79].

Within each sampling site, plant samples in the flowering stage were collected in the same soil sub-areas in spring (April 2022), placed in labeled plastic bags, and stored in a refrigerator at 4 °C until analysis. Both the aerial parts (stem, leaves, flowers) and roots from each plant were separated, washed in tap water, and then rinsed with distilled water. After washing, the roots were also sonicated in distilled water to ensure that all soil particles were removed. Fresh and dry (dried at 70 °C for 48 h in a drying and sterilizing oven) plant biomass was determined by weighing in a digital balance. The dried plant material was milled to powder homogenously and then digested with ultrapure nitric acid (69%) and hydrogen peroxide addition in a DigiPrep digestor. Multielemental concentration in the digested solution was determined by ICP-OES [73]. Quality assurance of chemical analysis in plants was performed using analytical blanks and certified reference material (NCSDC73348). The obtained results for the certified plant materials had a recovery rate between 86 and 103%, while procedural blanks were generally below the detection limit.

Moreover, since different color patterns of *C. transtagana* flower corollas were observed within the sampling sites, besides those described previously [59], the corolla type was visually determined and counted in all of the sampled plants.

### 3.4. Data Analysis

To compare the plant growth or enzymatic activity among the five sampling locations, a two-way ANOVA [(biomass parameters × locations) or (enzymes × locations)] was performed. Tukey’s HSD test (*p* < 0.05) was used to contrast the means. The tests were performed using the R studio version 1.2.1335 software.

To evaluate the plant behavior, the soil-to-plant transfer coefficient, translocation coefficient, and biological absorption coefficient of each element analyzed was calculated. The soil-to-plant transfer coefficient (TransferC = [total element in the shoots]/[total element in the soil]) illustrates the specific chemical elements accumulated in the shoots [80]. Plants with TransferC < 1 are considered nonaccumulators [81,82]. The translocation coefficient (TranslC = [total element in the shoots]/[total element in the roots]) assesses the plant capacity to translocate an element from the roots to the shoots [80].

The biological absorption coefficient (BAC = [total element in the roots]/[element in the available fraction of the soil]) evaluates the uptake capacity of an element by the plant. This latter coefficient can be divided into five groups considering the Perelman classification, which indicates the intensity of an element absorption by plant roots (intensive BAC, 10–100; strong BAC, 1–10; intermediate BAC, 1–0.1; weak BAC, 0.1–0.01; very weak BAC, 0.01–0.001) [83].

## 4. Conclusions

This preliminary study on the near-threatened *C. transtagana* species revealed that this species that thrives in metalliferous sites is tolerant to Cu and other PTEs, with different soil–plant transfer behaviors depending on the element analyzed. Since its historical areas of occurrence also appear to include non-metalliferous soils, comprehensive study should be carried out further to better understand whether the species thrives in such sites as well as other potential habitats and population sizes. Future work is needed to compare mine populations with other populations of the same species probably originated from non-contaminated soils.

This current study also showed that *C. transtagana* has a potential use in phytostabilization programs, considering its pioneer character on highly disturbed land, as in the studied soils, which contained very high total and available fractions of Cu content (up to 5.9 g/kg and 0.5 g/kg, respectively). It also would be interesting to assess the growth and elemental composition of the mine population and a control population from non-contaminated soils. This knowledge is important for planning an in situ conservation strategy for this species. Moreover, phytostabilization with native species is a low-cost technique avoiding ecological risks associated with the use of non-native species.

## Figures and Tables

**Figure 1 plants-13-02847-f001:**
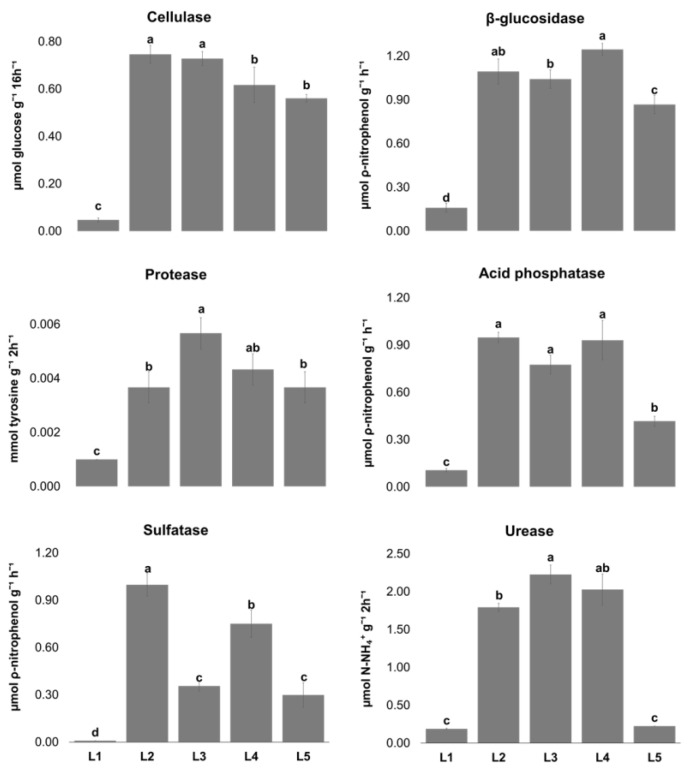
Enzymatic activity (mean ± sd; n = 3) in the five sampling locations (L1–L5) in the Aparis mine in Barrancos (southeast Portugal). For the same enzyme, locations (L1–L5) with different letters (a–d) are significantly different (two-way ANOVA [enzymes x location] followed by Tukey HSD test; *p* < 0.05).

**Figure 2 plants-13-02847-f002:**
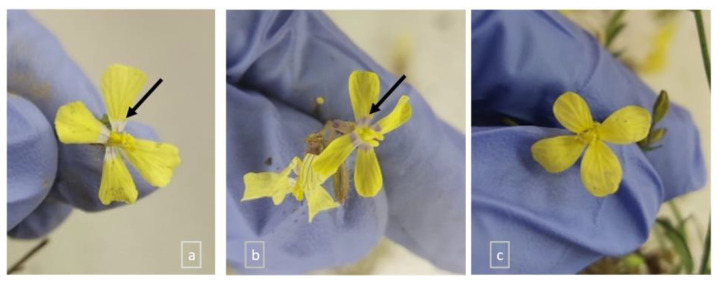
*Coincya transtagana* corolla patterns: corolla forming a white halo (indicated by an arrow) (**a**); a violet halo (indicated by an arrow) (**b**); and without halo (**c**).

**Figure 3 plants-13-02847-f003:**
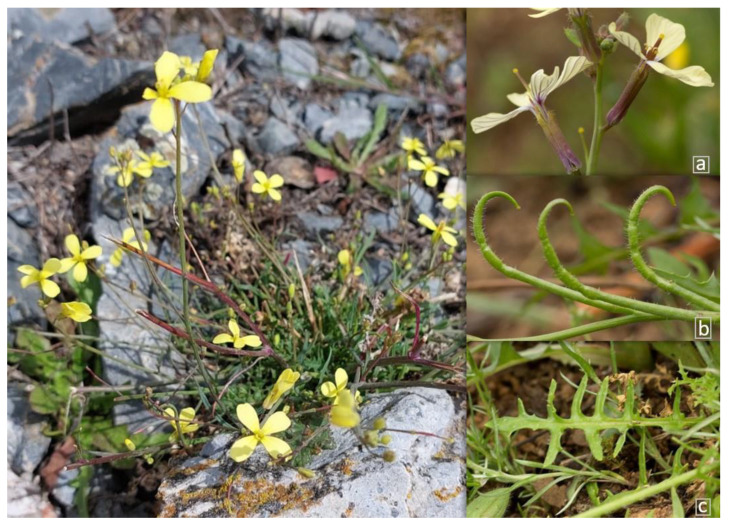
*Coincya transtagana* plant showing (**a**) white or pale-yellow petals and violet veins, (**b**) strongly curved and waxy fruits, and (**c**) basal leaves.

**Table 1 plants-13-02847-t001:** Chemical properties of the soil samples collected in five (L1–L5) sites in the Aparis mine in Barrancos (southeast Portugal). Each soil sample is a homogenized composite sample composed of three subsamples.

	L1	L2	L3	L4	L5
pH (H_2_O)	8.30	7.42	7.06	7.42	6.68
EC (mS/cm)	4.06	0.193	0.106	0.195	0.072
CaCO_3_ (g/kg)	28.1	2.50	4.20	4.10	3.00
C_org_ (g/kg)	5.17	25.91	32.68	25.96	17.32
N_total_ (g/kg)	0.37	2.81	2.51	2.79	2.47
N-NO_3_ (mg/kg)	2.20	13.00	5.25	15.40	4.20
P_extractable_ (mg/kg)	11.86	25.46	16.92	10.11	18.09
K_extractable_ (mg/kg)	58.76	108.53	124.75	180.72	82.52
Cation Exchange Capacity (CEC) and exchangeable cations (cmol_c_/kg)					
CEC	2.44	13.29	12.22	12.90	13.47
Ca	4.43	10.17	9.10	10.11	7.56
K	0.13	0.32	0.31	0.43	0.21
Mg	1.85	4.96	2.22	3.74	2.45
Na	<LD	0.04	<LD	0.07	<LD
Micronutrients (mg/kg)					
Cu	635.99	457.67	2024.94	647.56	1925.90
Fe	136.19	191.93	83.72	99.09	101.73
Mn	47.86	84.65	173.64	228.56	264.16
Zn	1.59	3.22	8.83	7.19	10.70
Macronutrients (g/kg)					
Ca	3.60	2.38	2.04	2.31	1.69
K	0.429	0.150	0.168	0.225	0.140
Mg	1.28	0.620	0.283	0.493	0.295
Na	0.010	0.016	0.017	0.027	0.011

**Table 2 plants-13-02847-t002:** Total and available fractions of the elements in the soils collected in the Aparis mine in Barrancos (southeast Portugal). DL: detection limit. Each soil sample is a homogenized composite sample composed of three subsamples.

Samples	As	Co	Cr	Cu	Mn	Ni	Pb	S	Zn
	Pseudototal (mg/kg)								
L1	92.7	13.7	14	1330	625	27.7	1.30	410	28.9
L2	37.9	21.3	24	1460	469	39.8	6.80	1110	58.6
L3	136	25.2	19	5940	755	39.1	12.20	1100	87.5
L4	60	25.7	23	2670	857	39.6	10.10	1320	79.4
L5	118	29	21	5720	1180	40.40	7.40	990	103
MAVs *	11	22	160	140	-	100	45	-	340
	Available fraction extracted with RHIZO solution (mg/kg)								
L1	4.1	1.32	0.3	220	19.9	1.22	<DL	30	0.93
L2	2.7	1.61	<0.2	48.8	31.8	1.12	<DL	20	1.36
L3	1.7	1.23	<0.2	502	62.8	0.83	<DL	50	5.26
L4	1.1	1.5	<0.2	160	70.5	0.85	<DL	30	3.57
L5	1.7	0.74	<0.2	491	69.6	0.81	<DL	30	5.50
%, ^ŧ^	1.4–7.1	2.6–9.6	≤2.1	3.3–16.5	3.2–8.3	2.0–4.4	-	1.8–7.3	2.3–6.0

* MAVs = maximum allowed values [26]. ^ŧ^ Percentage (minimum–maximum) of the available fraction of the element in relation to the pseudototal.

**Table 3 plants-13-02847-t003:** Total concentrations of the elements in the aerial part and roots of the *Coincya transtagana* lants collected in five (L1–L5) sites in the Aparis mine in Barrancos (southeast Portugal). DL—detection limit; concentrations of Ca, Fe, K, Mg, P, and S in g/kg dry weight; concentrations of As, Co, Cu, Cr, Mo, Ni, Pb, and Zn in mg/kg dry weight.

		L1	L2	L3	L4	L5
As	aerial part	28.74	<DL	4.74	<DL	<DL
	roots	10.93	<DL	<DL	<DL	<DL
Co	aerial part	4.39	0.96	0.71	1.01	0.57
	roots	0.73	0.50	0.77	0.63	0.61
Cu	aerial part	285.93	39.73	103.98	74.92	78.21
	roots	66.89	26.54	151.61	70.85	188.39
Cr	aerial part	39.33	10.68	10.67	10.14	4.25
	roots	6.07	5.03	4.59	2.51	6.08
Ni	aerial part	22.69	6.09	5.93	4.94	2.98
	roots	3.64	2.89	3.37	2.51	3.80
Pb	aerial part	3.03	<DL	1.19	1.27	<DL
	roots	<DL	<DL	<DL	<DL	<DL
Zn	aerial part	33.43	25.63	29.05	29.03	52.99
	roots	19.79	22.64	35.68	30.28	51.66
Ca	aerial part	23.6	20.72	14.70	14.45	21.39
	roots	6.798	5.447	5.988	7.462	6.320
Fe	aerial part	5.87	1.153	0.922	1.495	0.545
	roots	0.51	0.252	0.479	0.541	0.735
K	aerial part	14.21	15.16	12.57	12.07	12.84
	roots	13.23	17.61	13.83	16.58	13.52
Mg	aerial part	10.17	5.147	2.691	3.524	3.570
	roots	3.144	3.044	2.098	2.739	3.723
Mn	aerial part	149.77	27.77	30.83	44.37	34.00
	roots	16.99	10.06	21.44	26.38	36.46
Mo	aerial part	8.77	8.86	1.19	2.03	7.37
	roots	6.92	7.17	<DL	1.13	7.60
P	aerial part	2.18	3.91	2.88	3.37	3.26
	roots	1.42	3.32	2.43	2.09	3.25
S	aerial part	8.02	6.19	4.51	5.58	4.53
	roots	4.86	3.77	2.91	4.90	2.73

**Table 4 plants-13-02847-t004:** Translocation coefficient (TranslC), transfer coefficient (TransferC), and biological absorption coefficient (BAC) of As, Co, Cu, Cr, Mn, Mo, Ni, Pb, and Zn of *Coincya transtagana* at different sampling sites (L1–L5) in the Aparis mine in Barrancos (southeast Portugal).

		L1	L2	L3	L4	L5
	TranslC	2.63	<1.0	1.58	<1.0	<1.0
As	TransferC	1.62	<1.0	<1.0	<1.0	<1.0
	BAC	2.67	<1.0	<1.0	<1.0	<1.0
	TranslC	6.02	1.91	0.93	1.62	0.93
Co	TransferC	0.32	0.05	0.03	0.04	0.02
	BAC	0.55	0.31	0.62	0.42	0.82
	TranslC	4.28	1.50	0.69	1.06	0.42
Cu	TransferC	0.22	0.03	0.02	0.03	0.01
	BAC	0.30	0.54	0.30	0.44	0.38
	TranslC	6.48	2.12	2.32	4.04	0.70
Cr	TransferC	2.81	0.45	0.56	0.44	0.20
	BAC	20.23	<20.0	<20.0	<20.0	<20.0
	TranslC	8.81	2.76	1.44	1.68	0.93
Mn	TransferC	0.24	0.06	0.04	0.05	0.03
	BAC	0.85	0.32	0.34	0.37	0.52
	TranslC	1.27	1.24	<1.0	1.79	0.97
Mo	TransferC	10.45	14.77	1.77	3.50	9.69
	BAC	<0.15	<0.15	<0.15	<0.15	<0.15
	TranslC	6.23	2.10	1.76	1.97	0.78
Ni	TransferC	0.82	0.15	0.15	0.13	0.07
	BAC	2.99	2.58	4.06	2.96	4.69
	TranslC	<1.0	<1.0	<1.0	<1.0	<1.0
Pb	TransferC	2.32	<1.0	0.10	0.13	<1.0
	BAC	<5.0	<5.0	<5.0	<5.0	<5.0
	TranslC	1.69	1.13	0.81	0.96	1.03
Zn	TransferC	1.15	0.44	0.33	0.37	0.51
	BAC	21.28	16.65	6.78	8.48	9.39

## Data Availability

Data are contained within the article.

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
