# Peer review of "Unveiling the Behavior of an Endangered Facultative Cuprophyte Coincya Species in an Abandoned Copper Mine (Southeast Portugal)"

_plants, 2024, doi:10.3390/plants13202847_

Round 1
Reviewer 1 Report
Comments and Suggestions for Authors
Disclosure of a facultative cuprophyte Coincya species behavior in an abandoned copper mine in southeast Portugal
Ana Delaunay Caperta et alii
In this paper, they document mineral element concentrations in roots and aerial parts of a plant species (Coincya transtagana) on copper rich soil in S Portugal. The study species is original and has apparently not been previously studied for adaptation to soil.
The soil of the study site is characterised for chemical properties and enzymatic activities; however, it is not analysed for granulometry. Eight trace elements are considered, which is good.
The chemical analyses of soil and plant materials have apparently been carefully conducted.
However, I have major doubts as to the significance and importance of the results.
First, there are no testable hypotheses formulated in the introduction.
The only hypothesis is “the hypothesis of our study was that C. transtagana could be tolerant to high concentrations of potential hazardous elements (PHE) in metalliferous soils, particularly Cu.” It is actually not a testable hypothesis.
Their conclusion is that the species is indeed tolerant.
However, unfortunately, such a conclusion cannot be supported by their experimental design.
Metal tolerance can only be demonstrated if two conditions are fulfilled:
1. The species should be cultivated on a range of Cu concentration in the growth medium
2. The species/population must be compared to a control population /species.
Neither of these requirements is met here.
It is a pity that only one population was used. They should have compared the metalliferous population with a population from a non metalliferous site.
The fact that the species thrives on metalliferous soil of course suggests that it possesses some form a “tolerance” to the local soil, but this is trivial/tautological!
This could be due either to constitutive tolerance and phenotypic plasticity, or, alternatively, metalliferous populations could have evolved increased tolerance due to natural selection.
They have characterised soil and plants in 5 different locations in the study site. However, they do not use this sampling design to test hypotheses; they could have explored correlations between soil factors and plant traits (mineral element concentrations in tissue, and plant weight); this has not been done.
I have made many further comments on the manuscript.

Needs moderate editing
Author Response
Comments and Suggestions for Authors
Disclosure of a facultative cuprophyte Coincya species behavior in an abandoned copper mine in southeast Portugal
Ana Delaunay Caperta et alii
In this paper, they document mineral element concentrations in roots and aerial parts of a plant species (Coincya transtagana) on copper rich soil in S Portugal. The study species is original and has apparently not been previously studied for adaptation to soil.
The soil of the study site is characterised for chemical properties and enzymatic activities; however, it is not analysed for granulometry. Eight trace elements are considered, which is good.
The chemical analyses of soil and plant materials have apparently been carefully conducted.
However, I have major doubts as to the significance and importance of the results.
First, there are no testable hypotheses formulated in the introduction.
The only hypothesis is “the hypothesis of our study was that C. transtagana could be tolerant to high concentrations of potential hazardous elements (PHE) in metalliferous soils, particularly Cu.” It is actually not a testable hypothesis.
Their conclusion is that the species is indeed tolerant.
However, unfortunately, such a conclusion cannot be supported by their experimental design.
Metal tolerance can only be demonstrated if two conditions are fulfilled:
- The species should be cultivated on a range of Cu concentration in the growth medium
- The species/population must be compared to a control population /species.
Neither of these requirements is met here.
It is a pity that only one population was used. They should have compared the metalliferous population with a population from a non metalliferous site.
The fact that the species thrives on metalliferous soil of course suggests that it possesses some form a “tolerance” to the local soil, but this is trivial/tautological!
This could be due either to constitutive tolerance and phenotypic plasticity, or, alternatively, metalliferous populations could have evolved increased tolerance due to natural selection.
They have characterised soil and plants in 5 different locations in the study site. However, they do not use this sampling design to test hypotheses; they could have explored correlations between soil factors and plant traits (mineral element concentrations in tissue, and plant weight); this has not been done.
I have made many further comments on the manuscript.
- We thank the Reviewer for his/her critical insights that helped improving the manuscript. We reworded the manuscript and tried to clarify unclear or misunderstanding sentences.
We provided a revised version of the manuscript with track-changes and a cleaned version.
- - “they could have explored correlations between soil factors and plant traits (mineral element concentrations in tissue, and plant weight); this has not been done.”
We think we explored correlations between soil factors and plant traits. Although a coarse-scale analysis, this work constitutes the first soil-plant interaction study for this endangered species providing the basis for subsequent fine-scale studies in selected natural populations.
As the levels of Cu in the plant are among those considered excessive or toxic (> 20-30 mg/kg) some plants can develop tolerance to high levels of the element by various mechanisms. Both Mengel & Kirkby (2001) and Sravistava & Gupta (1996) refer to tolerant species.
We don't think it's necessary to carry out lab tests to see if the plants are tolerant. The tests would allow us to assess up to what concentration of available Cu (these could be hydroponics tests or, if it's in soil, how much Cu is available in the soil) the plant could withstand without dying or showing signs of toxicity (it could even be inhibition of growth or death).
On the other hand, since plants cannot be translocated and seed germination experiments were not yet carried out, seedlings for metal tolerance studies were not obtained. However, we think that this suggestion is very useful and a metal tolerance approach like that suggested for the Reviewer could be conducted in the future.
3 – “how did you assess CEC and calcium in soil with so much calcium carbonate?”
4- "it is not possible to determine copper at such a high precision with your methods. Truncate to 3-4 digits as in table 2”.
5- “The multielemental (pseudototal) soils composition revealed that the PHE (Table 2) were below the maximum allowed values (MAV) for
agricultural use [26],”
We agree with the Reviewer. The sentence was removed from the text.
6- “I do not understand this sentence. … It is not related to our hypothesis”
We agree that the sentence is inappropriate in the study context. Therefore, we removed it.
7- All sentences comparing the obtained values with plants with agricultural uses were removed from the text.
Done.
- - “Needs moderate editing”
English was revised throughout the manuscript. We are uploading a track changes document with the new wording with the addition of new information.

Reviewer 2 Report
Comments and Suggestions for Authors
The paper “"Disclosure of a facultative cuprophyte Coincya species behavior in an abandoned copper mine in southeast Portugal" provides valuable insights into the behaviour of Coincya transtagana in metalliferous soils, evaluation its potential for phytoremediation. However, the paper would benefit from clearer articulation of its hypothesis, more comprehensive literature integration, detailed methodological descriptions, and a more critical discussion of its findings.
Here are the potential gaps and suggestions for improvements:
The hypothesis is not clearly stated in the introduction. While the goals are mentioned, the linkage between the hypothesis and the specific research objectives could be better articulated. Clearly define the hypothesis in the introduction and ensure that the objectives are directly linked to this hypothesis. This will help in guiding the reader through the study's rationale.
The literature review does not provide a comprehensive background on the metallophytes, specifically in the context of Coincya transtagana. There is also limited discussion on previous studies conducted on similar species in analogous environments. Expand the literature review to include a broader discussion on metallophytes and previous research on Coincya species, especially focusing on their behavior in metal-rich soils. This would contextualize the study better and highlight its novelty.
The methods section provides a basic overview of the procedures but lacks detailed descriptions of certain protocols, especially regarding soil and plant analysis techniques. The selection criteria for the sampling sites and the justification for the chosen methodologies are also not well-explained. Include detailed protocols for all experimental procedures, including specific steps for soil and plant analysis. Additionally, provide a rationale for the choice of sampling sites and methodologies to strengthen the study's validity.
Some of the data interpretations, particularly regarding the enzymatic activities and plant-metal interactions, are superficial. The discussion could benefit from a more in-depth analysis of why certain results were observed, considering both biological and environmental factors. Enhance the discussion by linking the observed data to existing theories and research. Consider alternative explanations for unexpected results and discuss potential confounding factors that might have influenced the findings.
While statistical tests are mentioned, the choice of these tests is not justified, and there is limited discussion on the assumptions underlying these tests. Furthermore, the paper does not explore the possibility of other statistical models that could be more appropriate. Justify the choice of statistical tests and ensure that the assumptions of these tests are clearly stated. Consider exploring more advanced statistical models or multivariate analyses to provide a more comprehensive understanding of the data.
Improve figure and table captions to be more descriptive. Consider adding more figures, such as graphical representations of key findings, to make the data more accessible and easier to interpret.
The discussion section tends to reiterate the results rather than providing a critical analysis of their implications. There is also a lack of a clear connection between the study's findings and broader ecological or conservation implications. Focus on critically analyzing the results in the discussion, addressing how the findings contribute to the current understanding of metallophyte ecology. The conclusion should clearly state the study's contributions and suggest directions for future research
Comments on the Quality of English LanguageModerate English is required.
Author Response
Comments and Suggestions for Authors
The paper “"Disclosure of a facultative cuprophyte Coincya species behavior in an abandoned copper mine in southeast Portugal" provides valuable insights into the behaviour of Coincya transtagana in metalliferous soils, evaluation its potential for phytoremediation. However, the paper would benefit from clearer articulation of its hypothesis, more comprehensive literature integration, detailed methodological descriptions, and a more critical discussion of its findings.
Here are the potential gaps and suggestions for improvements:
1 - The hypothesis is not clearly stated in the introduction. While the goals are mentioned, the linkage between the hypothesis and the specific research objectives could be better articulated. Clearly define the hypothesis in the introduction and ensure that the objectives are directly linked to this hypothesis. This will help in guiding the reader through the study's rationale.
We appreciate the Reviewer’s constructive and helpful feedback.
We have made significant revisions throughout the manuscript to enhance clarity and improve the overall readability for the audience.
We provided a track changes version of the manuscript, so that the Reviewer can appreciate the changes made.
2 - The literature review does not provide a comprehensive background on the metallophytes, specifically in the context of Coincya transtagana. There is also limited discussion on previous studies conducted on similar species in analogous environments. Expand the literature review to include a broader discussion on metallophytes and previous research on Coincya species, especially focusing on their behavior in metal-rich soils. This would contextualize the study better and highlight its novelty.
We expanded the Literature Review as well as improved the Discussion section to include more information on metallophytes as well as on other Coincya species.
The methods section provides a basic overview of the procedures but lacks detailed descriptions of certain protocols, especially regarding soil and plant analysis techniques. The selection criteria for the sampling sites are also not well-explained. Include detailed protocols for all experimental procedures, including specific steps for soil and plant analysis. Additionally, provide a rationale for the choice of sampling sites and methodologies to strengthen the study's validity.
As requested by the Reviewer we provided detailed descriptions of methods. These are classic methodologies used in the characterisation of soils and plants and used in various studies using soils and plants from contaminated areas, and in similar works published in the international literature. We also incorporated justifications for the chosen methodologies, sampling sites and plants (below).
3 - Some of the data interpretations, particularly regarding the enzymatic activities and plant-metal interactions, are superficial. The discussion could benefit from a more in-depth analysis of why certain results were observed, considering both biological and environmental factors. Enhance the discussion by linking the observed data to existing theories and research. Consider alternative explanations for unexpected results and discuss potential confounding factors that might have influenced the findings.
We deepened the Discussion section regarding the enzymatic activities.
4- While statistical tests are mentioned, the choice of these tests is not justified, and there is limited discussion on the assumptions underlying these tests. Furthermore, the paper does not explore the possibility of other statistical models that could be more appropriate. Justify the choice of statistical tests and ensure that the assumptions of these tests are clearly stated. Consider exploring more advanced statistical models or multivariate analyses to provide a more comprehensive understanding of the data.
We understand the Reviewer claims on other statistical models to be used.
We reworded the manuscript to make clear that each of our samples are already an average of samples (technical replicates) as mentioned in the M&Ms. For both soil and plant samples (L1 to L5) these correspond to composite samples. Regarding soils, this is a mixture of three sub-samples per area (L1-L5) totalizing ~3 kg/area. As for plants, a composite sample of around 18-20 plants per site collected from the three sub-areas/sites where the soils were collected (~6 plants per sub-area) was made. The homogenisation of the total sample (3 kg of soil and 18-20 plants per site) was guaranteed, so the result of this homogenisation is already an average of three soil sub-samples per site and 18-20 plants/site. Therefore, the data on Table 1&2 cannot be presented as means ± STD.
For some of the data as e.g. dehydrogenase (two replicas per site). we don’t have enough repetitions for statistical tests.
5 - Improve figure and table captions to be more descriptive. Consider adding more figures, such as graphical representations of key findings, to make the data more accessible and easier to interpret.
Table captions were improved and provided in the new version of the manuscript.
6 - The discussion section tends to reiterate the results rather than providing a critical analysis of their implications. There is also a lack of a clear connection between the study's findings and broader ecological or conservation implications. Focus on critically analyzing the results in the discussion, addressing how the findings contribute to the current understanding of metallophyte ecology. The conclusion should clearly state the study's contributions and suggest directions for future research.
Plant-interaction studies are essential for preserving endangered species. We rewrote the Discussion section to accommodate the Reviewer observations.

Reviewer 3 Report
Comments and Suggestions for Authors 1、The conclusion is too short, please summarise the research done in detail in the text. 2、The overall layout of the article needs a little more adjustment, there are many blank areas. 3、The format of individual reference citations needs to be modified. 4、Some of the references are old, and new ones can be added as appropriate. Comments on the Quality of English Language The essay's grammatical structure and use of vocabulary is accurate and clearly expresses the content of the essay.Author Response
Comments and Suggestions for Authors
We thank the Reviewer for the suggestion to improve the manuscript.
1、The conclusion is too short, please summarise the research done in detail in the text.
The Conclusions section was rewritten.
2、The overall layout of the article needs a little more adjustment, there are many blank areas.
We agree with the Reviewer. The blank areas will be removed after proper editorial adjustments.
3、The format of individual reference citations needs to be modified.
The References were revised.
4、Some of the references are old, and new ones can be added as appropriate.
Some references were added, others were deleted.
Comments on the Quality of English Language
The essay's grammatical structure and use of vocabulary is accurate and clearly expresses the content of the essay.
Thank you!

Reviewer 4 Report
Comments and Suggestions for Authors
1. L41-42, the text above talks about Cu contamination, how come in this sentence Co exists? And which element do the threshold values in the bracket refer to?
2. Data in Table 1&2 should be presented as means ± STD.
3. L127 4,1 mg/kg should be 4.1 mg/kg. Same as L152.
4. How come the methods and materials section is after the result?
5. L153 and Fig. 1, where is the data for dehydrogenase activity?
6. Figure 2, where is the arrow indicating the different halos?
7. Table 3 should give the normal plant limit concentrations for these metals. Table 3 and 4 are suggested to be combined. Table 4, can’t BAC for Cr be determined more accurately?
8. L251, biological absorption concentration is determined as CAB or BAC?
9. L281, the semicolon before ‘However’ should be a full stop.
10. Why do the abstract and conclusion have to emphasize Cu? Some other elements obviously show regular absorption behavior, such as Mo.
Comments on the Quality of English Languagemoderate edition is needed.
Author Response
Comments and Suggestions for Authors
We thank the Reviewer for its comments which have helped to improve the manuscript quality.
- L41-42, the text above talks about Cu contamination, how come in this sentence Co exists? And which element do the threshold values in the bracket refer to?
Although Aparis is a copper mine it also has other elements associated with the mineralisation (ore), as happens in both Ossa Morena Zone and PIB mines (Portugal and Spain). In three of the Aparis soils, Co values exceeded the MAV values, and in the other two they are at the MAV limit.
Regarding to the threshold values in brackets, the values 1-> 25 g/kg refer to potentially toxic elements as Sb, As, Cd, Cu, Pb, Hg, W and Zn generally existent in mining areas. For example, a Cu mine as Aparis has very high levels of this element.
- Data in Table 1&2 should be presented as means ± STD.
We reworded the manuscript to make clear that our samples are already an average of samples (technical replicates) as mentioned in the M&Ms. For both soil and plant samples (L1 to L5) these correspond to composite samples. Regarding soils, this is a mixture of three sub-samples per area (L1-L5) totalizing ~3 kg/area. As for plants, a composite sample of around 18-20 plants per site collected from the three sub-areas/sites where the soils were collected (~6 plants per sub-area) was made. The homogenisation of the total sample (3 kg of soil and 18-20 plants per site) was guaranteed, so the result of this homogenisation is already an average of three soil sub-samples per site and 18-20 plants/site.
Therefore, the data on Table 1&2 cannot be presented as means ± STD.
- L127 4,1 mg/kg should be 4.1 mg/kg. Same as L152.
Done.
- How come the methods and materials section is after the result?
This is the format the journal adopted.
- L153 and Fig. 1, where is the data for dehydrogenase activity?
The dehydrogenase activity cannot be represented in graphical form since by contrast with other enzymes, we only have two replicas per site.
- Figure 2, where is the arrow indicating the different halos?
The new figure has the halos arrowed.
- Table 3 should give the normal plant limit concentrations for these metals. Table 3 and 4 are suggested to be combined. Table 4, can’t BAC for Cr be determined more accurately?
As the Cr values in the available fraction of the soil (Table 2) are lower than the Detection Limit and that after converting mg/L in the device's reading solution (20 μg/L, 73. Activation Laboratories. Aqua Regia “Partial” Digestion. Available online: https://actlabs.com/geochemistry/exploration-geochemistry/aqua-regia-partial-digestion/ (accessed: 01-04-2024)) to the mass of soil used gave a value <0.2 mg/kg. This value was used for the calculation and as is evident in Table 2, there is not an exact value but < 20.
- L251, biological absorption concentration is determined as CAB or BAC?
Corrected.
- L281, the semicolon before ‘However’ should be a full stop.
Corrected.
- Why do the abstract and conclusion have to emphasize Cu? Some other elements obviously show regular absorption behavior, such as Mo.
We reworded the Abstract and the Conclusions.
Comments on the Quality of English Language
moderate edition is needed.
English was revised.

Round 2
Reviewer 1 Report
Comments and Suggestions for Authors
The conclusion should be amended, to acknowledge the methodological limitations of the paper. In particular, they should make clear that future work is needed, in experimental conditions, in order to:
- compare the mine population with other populations of the same species originating from non contaminated soil;
- assess growth and mineral element composition of the mine population and a control population from non contaminated soil, in response to a gradient of Cu (and other metals).
Author Response
The conclusion should be amended, to acknowledge the methodological limitations of the paper. In particular, they should make clear that future work is needed, in experimental conditions, in order to:
- compare the mine population with other populations of the same species originating from non contaminated soil;
- assess growth and mineral element composition of the mine population and a control population from non contaminated soil, in response to a gradient of Cu (and other metals).
The Conclusion was amended to adapt the claims of the Reviewer.
Some edits were realized throughout the manuscript.
Reviewer 2 Report
Comments and Suggestions for Authors/
Comments on the Quality of English Languageminor editing
Author Response
minor editing
We did minor English editing.
Reviewer 4 Report
Comments and Suggestions for Authors
If a value can be presented as a mean, it should have the information of the differences between the parallel samples and STD is a must to give.
Author Response
If a value can be presented as a mean, it should have the information of the differences between the parallel samples and STD is a must to give.
If a value can be presented as a mean, it should have the information of the differences between the parallel samples and STD is a must to give.
Since technical replicates are a good laboratory practice in determination of a particular parameter to be analysed. Aliquots (2 or 3) are taken from the same sample (parts of the same sample, not different samples) to determine a parameter to gauge analytical quality. We did it.
Nonetheless, these technical replicates are not valid for statistical treatment. To this end, we would need at least three samples from different soils in the same site.
As answered previously to the Reviewer, the values in tables 1 and 2 are not presented in the manuscript as means. Contrary to what the reviewer wrote ‘If a value can be presented as a mean....’, for the values included in Tables it is not possible to give information on the standard deviation.